# The Influence of Hydrodynamic Conditions in a Laboratory-Scale Bioreactor on *Pseudomonas aeruginosa* Metabolite Production

**DOI:** 10.3390/microorganisms11010088

**Published:** 2022-12-29

**Authors:** Maciej Konopacki, Joanna Jabłońska, Kamila Dubrowska, Adrian Augustyniak, Bartłomiej Grygorcewicz, Marta Gliźniewicz, Emil Wróblewski, Marian Kordas, Barbara Dołęgowska, Rafał Rakoczy

**Affiliations:** 1Department of Chemical and Process Engineering, Faculty of Chemical Technology and Engineering, West Pomeranian University of Technology in Szczecin, Piastów Avenue 42, 71-065 Szczecin, Poland; 2Department of Laboratory Medicine, Chair of Microbiology, Immunology and Laboratory Medicine, Pomeranian Medical University in Szczecin, Powstańców Wielkopolskich 72, 70-111 Szczecin, Poland; 3Chair of Building Materials and Construction Chemistry, Technische Universität Berlin, Gustav-Meyer-Allee 25, 13355 Berlin, Germany; 4Institute of Biology, University of Szczecin, Wąska 13 Str., 71-415 Szczecin, Poland

**Keywords:** bacterial biomass, pyocyanin, rhamnolipids, bioprocessing, mixing, aeration

## Abstract

**Highlights:**

This study demonstrated that the process production of *Pseudomonas aeruginosa* metabolites might be successfully carried out by using a batch bioreactor. This process is dependent on the hydrodynamic conditions. Therefore, we tested and analyzed the fluid behaviour for gassed and ungassed conditions with the application of the various types of hydrodynamic criteria (e.g., power consumption, mixing time, mixing energy, the volumetric gas-liquid mass transfer coefficient). The obtained hydrodynamic parameters might be successfully used in the bioprocessing of the different substances with the application of the tested bioreactor.The growth and metabolite production of *Pseudomonas aeruginosa* was analyzed under selected hydrodynamic conditions. We found that the highest oxygen mass transfer is connected with the most intensive pyocyanin production. The rhamnolipids were produced under low mixing conditions and moderate oxygen mass transfer.The obtained data showed that the process production of *Pseudomonas aeruginosa* metabolites is possible with the use of a batch bioreactor. It should be emphasized that the hydrodynamic conditions in the used mixing system have strongly influenced the obtained product. Therefore, the hydrodynamic analysis for bioprocessing or process production of bioproducts with the application of the mixing systems is required.

**Abstract:**

Hydrodynamic conditions are critical in bioprocessing because they influence oxygen availability for cultured cells. Processes in typical laboratory bioreactors need optimization of these conditions using mixing and aeration control to obtain high production of the desired bioproduct. It could be done by experiments supported by computational fluid dynamics (CFD) modeling. In this work, we characterized parameters such as mixing time, power consumption and mass transfer in a 2 L bioreactor. Based on the obtained results, we chose a set of nine process parameters to test the hydrodynamic impact on a selected bioprocess (mixing in the range of 0–160 rpm and aeration in the range of 0–250 ccm). Therefore, we conducted experiments with *P. aeruginosa* culture and assessed how various hydrodynamic conditions influenced biomass, pyocyanin and rhamnolipid production. We found that a relatively high mass transfer of oxygen (k_L_a = 0.0013 s^−1^) connected with intensive mixing (160 rpm) leads to the highest output of pyocyanin production. In contrast, rhamnolipid production reached maximal efficiency under moderate oxygen mass transfer (k_L_a = 0.0005 s^−1^) and less intense mixing (in the range of 0–60 rpm). The results indicate that manipulating hydrodynamics inside the bioreactor allows control of the process and may lead to a change in the metabolites produced by bacterial cells.

## 1. Introduction

Culturing conditions play an essential role in biotechnological processes based on microbial production. Depending on the preferences of the inoculated microorganism and the acquired product, the process may be performed under aerobic, anoxic or anaerobic conditions. Many biotechnological processes require oxygen, a vital component of oxidative phosphorylation in bacteria and fungi, as an efficient electron acceptor [1,2,3,4,5]. Playing a crucial role in energy production in cells, the availability of this gas is expressed as the oxygen transfer rate (OTR) and is often listed as the limiting factor in bioprocessing due to the low oxygen solubility in the liquid [6]. OTR is a product of the oxygen concentration gradient and volumetric mass transfer coefficient (k_L_a) [7], while k_L_a consists of the mass transfer coefficient (k_L_) and gas-liquid area exposed to transfer in a liquid volume (a). Therefore, increasing k_L_a provides higher oxygen availability to the growing biomass. It can be achieved by altering bioreactor construction and operating parameters [8]. Parameters such as the geometry of the bioreactor, impeller/turbine type, aeration, and stirring rate also influence k_L_a [9]. Therefore, the study of hydrodynamics allowing the determination of k_L_a and OTR enables finding the optimal conditions for the carried-out process. Moreover, hydrodynamic experimental studies are often supported with mathematical modeling based on a computational fluid dynamics approach that helps understand ongoing bioreactor processes [10]. Many biotechnological processes are performed in universal bioreactors, such as the UniVessel^®^ Glass bioreactor (Sartorius, Germany), that provide stirring and aeration rate control and O_2_, CO_2_, and pH level monitoring. Such bioreactors or similar ones were previously employed in different bioprocesses for biomass and metabolite production, including *Penicillium* fermentation [11], mammalian cell production [12,13] and *Clostridium kluyveri* cultivation [14].

Recently, there has been growing interest in utilizing *Pseudomonas aeruginosa* for biotechnological processes, as it is a recognized producer of various utile metabolites, including pyocyanin [15,16], rhamnolipids [17], hydroxy fatty acids [18], polyhydroxybutyrate [19], alkaline lipase [20,21] and exopolysaccharide (alginate) [22]. The current study focuses on the first two of these metabolites. Although it is an opportunistic pathogen, it is an exclusive producer of pyocyanin. Pyocyanin is a bacterial pigment that is recognized as a virulence factor. Nevertheless, it may find possible use in many industries. Pyocyanin potential was reported in the case of cancer and mitochondrial disease treatment [23,24,25,26], energy production in microbial fuel cells [27,28,29,30], and phage induction [31] and has even been suggested as a food preservative (however, these reports have limitations concerning pyocyanin potential toxicity) due to its broad antimicrobial activity [32]. The considerable potential of pyocyanin was found in inhibiting plant pathogens, where 150 and 200 ppm concentrations of pyocyanin were found to be effective against *Magnaporthe grisea* and *Xanthomonas oryzae*, respectively [33]. On the other hand, rhamnolipids are surfactants that possess unique properties, such as low toxicity against eukaryotic cells, high biodegradability, tolerance to a wide range of conditions (pH, temperature, or salinity) and a relatively low critical micelle concentration (CMC) [34]. These amphiphilic molecules can act as emulsifiers, detergents, wetting and foaming agents and dispersants [35]. Due to these properties, they can be widely used in industry [36], agriculture as surfactants in pesticide formulas and plant immune stimulation agents [37,38,39], environmental protection for bioremediation of contaminated soil, petrochemistry in microbial enhanced oil recovery [40], the food industry as additives [41,42] and the cosmetic industry and pharmacy as detergents and preservatives [43,44,45,46]. Rhamnolipids also are *Pseudomonas* metabolites that are industrially produced. Finally, the production of *P. aeruginosa* biomass can propagate bacteriophages that can help combat multidrug-resistant infections, e.g., in burn wards and in patients suffering from cystic fibrosis [47,48,49]. *P. aeruginosa* is a facultative anaerobe that prefers oxygen. However, it can also use nitrogen oxides as alternative terminal electron acceptors, which means that anoxic conditions in a bioreactor can lead to metabolic changes in a part of the growing bacterial population. This potent biofilm maker is also heavily resistant to harsh environmental conditions (temperature, pH, oxidative stress, other microorganisms) thanks to a range of genes responsible for its adaptability (efflux pumps, enzymes, pigments, surfactants, pyocins, etc.). High tolerance to physical and chemical factors makes the production of desirable metabolites in this microorganism demanding, and every process requires additional optimization. It is particularly significant in the case of pyocyanin production that is, thus far, scarcely described in the literature (based on a Scopus database search).

Therefore, in this work, we evaluated the effects of hydrodynamic conditions, i.e., stirring and aeration, on the laboratory-scale production (2 L working volume) of biomass, rhamnolipid and pyocyanin by *Pseudomonas aeruginosa*.

## 2. Materials and Methods

### 2.1. Experimental Setup

In the current study, a UniVessel^®^ Glass 2 L (Sartorius, Germany) bioreactor was employed. A view and scheme of the bioreactor are presented in Figure 1.

The bioreactor setup consisted of a glass vessel (1) with a 2 L working volume equipped with an internal cooler (2), external heater (3), gas distributor (4), temperature sensor (5) and mechanical agitator with four vertical rectangular baffles (6). The agitator was powered by an electrical engine controlled by a bioreactor central unit that could be decoupled to connect the impeller shaft (7) with other equipment. On the agitator shaft, arranged on a central axis, two Rushton-type turbines (8) with six rectangular blades each (one near the vessel bottom, just above the gas distributor, and the second one at the middle of the vessel height) were mounted. The bioreactor system was also equipped with control probes-pH and dissolved oxygen (DO)-and peristaltic pumps that could add some substances to the bioreactor vessel through a designated injection port (9) placed in the top cover. All equipment could be controlled by the bioreactor central unit connected to a PC with designated software recording the measurements. This kind of bioreactor requires ex situ sterilization by autoclaving.

### 2.2. Hydrodynamic Conditions, Mixing Efficiency, and Mass Transfer Characteristics for the Tested Bioreactor

The hydrodynamic conditions in the tested bioreactor were defined using mixing time analysis and power consumption measurements. The mixing time was measured using a tracer method based on the pH changes within the process medium [8]. The bioreactor was equipped with two pH sensors connected to the multiparameter reader CX-601 (Elmetron, Zabrze, Poland) and placed in the bottom-right and top-left parts of the bioreactor vessel at the highest possible distance between the probes. The mixing time was evaluated for various impeller speeds ranging from 50 to 300 rpm. For each experiment, 50 mL of tracer (1 M aqueous solutions of citric acid) was introduced by a peristaltic pump into the vessel containing 2 L of conditioned water at the top of the liquid surface by the injection port through the bioreactors. The mixing time was determined as the time needed to reach 95% homogenization in a medium. All experiments were conducted in triplicate and included variants with and without additional aeration.

The power consumption was determined as the amount of energy transferred by the impeller to the mixed medium during the mixing process. This parameter was recorded by torque measurements with a VK 600 Control VISCOCLICK (IKA, Germany) connected directly to the impeller shaft. The energy input was calculated from the difference between impeller torque in the filled and empty vessel for the tested speed (in the range of 50 to 300 rpm).

The mass transfer in the bioreactor was evaluated for impeller speeds ranging from 50 to 300 rpm and under gas flow rates (GFRs) set from 125 to 1000 ccm. A dissolved oxygen sensor measured the amount of oxygen from compressed air. Residual oxygen was removed from the working volume by nitrogen before each experiment. The process was conducted until a steady gas saturation was reached in the liquid. All measurements were recorded three times.

### 2.3. Growth and Viability Measurements

The growth of the culture was monitored by optical density measurements (λ = 600 nm) for 48 h of incubation. The measurements were conducted every 2 h for the first 12 h and then at two single time points at 24 and 48 h. Moreover, an additional experiment was conducted to link the optical density of the cultures with the obtained dry biomass and the number of viable cells in the total plate count method. The viability of the cultures was measured employing the resazurin assay, which allows monitoring of cellular respiration. More comprehensive descriptions of these methods are provided in the Appendix A

### 2.4. Pyocyanin and Rhamnolipid Quantification

Based on spectrophotometric measurements, pyocyanin and rhamnolipid production were assessed after 48 h of incubation. Pyocyanin was extracted employing chloroform-hydrochloric acid methodology previously described elsewhere [50]. The rhamnolipid extraction assay was based on the chloroform-methylene blue methodology described by Pinzon and Ju [51]. The concentrations of pyocyanin and rhamnolipids were calculated according to the standard calibration curves shown in the Appendix A.

## 3. Results and Discussion

The obtained results connected with the tested bioreactor’s power consumption and mixing time are given in Appendix A, respectively. These materials contain results for the ungassed and gassed conditions. Additionally, the mass transport evaluation based on the concentration of dissolved oxygen in the bioreactor medium is given in Appendix A

The mixing energy might be calculated based on the obtained results of power consumption and mixing time. This parameter can connect the power consumption and the mixing time to evaluate the mixing efficiency within the bioreactor for various impeller speeds, both for ungassed and gassed conditions:(1)$$\begin{array}{l} E\propto n\;\Rightarrow \;\left(P\;\tau_{95}\right)\propto n\\ E_{g}\propto n\;\Rightarrow \;\left(P_{g}\;\tau_{95}\right)\propto n \end{array}$$
where *E* is the mixing energy for ungassed conditions, J, and *E_g_* is the mixing energy for gassed conditions, J. The results of mixing energy calculations are presented in Figure 2.

The mixing energy can determine the costs of the process and allow the comparison of the various mixing systems. The mixing energy is raised for every studied case with increasing impeller speed. Additional aeration significantly decreases the mixing energy, thus improving the process (the same effect can be achieved with a lower energetic cost). In the current studies, the mixing energy was improved five times for the highest analyzed gas flow rate, regardless of the impeller speed. This effect is strongly dependent on the gas flow rate in the bioreactor.

From a practical point of view, the mixing process might be evaluated using the optimization criterion concerning the minimum mixing work [52]. This criterion is expressed as follows:(2)$$\Pi_{1}=f\left(\Pi_{2}\right)\;\Rightarrow \;\left(\frac{PD\rho^{2}}{\nu^{3}}\right)=f\left(\frac{\nu \;\tau_{95}}{D^{2}}\right)$$
where: *P*–power consumption, J·s^−1^; *D*–diameter of the bioreactor, m; *ρ*–density, kg·m^−3^; ν–viscosity, m^2^·s^−1^.

The graphical representation of Equation (2) is shown in Figure 3 for the applied bioreactor, which in a particular range of the dimensionless number Π_2_ exhibits the lowest Π_1_ values (requires the lowest stirrer power in this range). It should also be noted that this graph allowed us to obtain the required mixing time, τ_95_, (proportional to Π_2_), with the lowest power, P, (proportional to Π_1_) and hence the minimum mixing work (Pτ_95_ = minimum). These results prove that the increase in the gas flow in the bioreactor causes a reduction in the mixing time (lower values of Π_2_). Moreover, the increase in the gas flow is associated with the reduction of power consumption (lower values of the number Π_1_).

The relationship between the numbers Π_1_ and Π_2_ shown in Figure 3, in addition to optimizing the mixing process (selecting the conditions under which we achieve the minimum mixing work), also allows us to determine the range of hydrodynamic conditions in the tested setups. It was decided that the production of metabolites by *Pseudomonas aeruginosa* would be tested for the hydrodynamic conditions given by the maximal, minimal and average number of Π_1_. Moreover, for each Π_1_ value, three various Π2 numbers were selected, corresponding to a specific gas flow rate, resulting in the configuration of 9 different process conditions for analysis. We decided to test gas flow rates in the range 0–4.17·10^−6^ m^3^·s^−1^ because further increment of gas flow created a very high disturbance in the bioreactor volume, which was observed during the hydrodynamic studies.

Additionally, CFD (computational fluid dynamics) simulations supported studies of bioreactor hydrodynamics to analyze the fluid velocity profiles and other velocity-related parameters. The tested bioreactor has two Rushton-type turbines mounted on a single shaft and four rectangular baffles. CFD analysis (Appendix A) shows that the impeller at low speed (Re_mix_ = 2696) created many dead zones (with poor mixing efficiency), especially in the upper part of the vessel and between impellers. A higher impeller speed (Re_mix_ = 7189) exhibits improved fluid circulation, which covers the total height of the vessel. For both cases, the best mixing can be observed around the gas distributor near the bottom. It is essential in the case of aeration, highly improving the oxygen mass transfer within the vessel.

Moreover, the presence of additional equipment such as an internal cooler, temperature sensor casing and gas pipe act as additional baffles disrupting the typical radial fluid motion created commonly by the Rushton turbine, especially for higher impeller speeds. It could be noted that the internal cooler or temperature sensor can create deflection for the fluid streamlines, therefore creating dead zones, a negative phenomenon in the case of mass transfer. Moreover, the fluid circulation inside the vessel is asymmetric, while a symmetric fluid flow pattern characterizes the typical geometry of the Rushton impeller mixer with baffles. It creates a situation where bioreactor geometry is hard to compare with other typical structures, making the study of hydrodynamics more critical for a new type of bioreactor. It should be highlighted that a high-speed Rushton turbine, which can create efficient mixing in the studied geometry, should not be a preferred option for bioprocessing due to the high shear stresses created by the impeller blade, which is harmful to cultured cells [53,54]. High shear stress induced by the impeller’s blade may induce changes in the metabolism or morphology of cells, leading even to lysis, and therefore should be avoided by reducing the impeller speed or optimizing its geometry [55].

The growth curves of *P. aeruginosa* cells were determined as changes in the optical density versus time. A typical growth curve is illustrated in Figure 4a.

Experimental data points were approximated using a sigmoidal-shaped mathematical function described previously [56]. Its formula allowed us to estimate a few crucial parameters, such as the maximum specific growth rate and lag duration. Moreover, a growth factor combining biomass production and growth kinetics was calculated [56]. Estimated bacterial growth parameters are listed in Appendix A. The results of bacterial growth in the form of growth factors under various process conditions (impeller speed and airflow rate), represented by corresponding Reynolds numbers, are presented in Figure 4b.

Both impeller speed and airflow rate had a high impact on bacterial growth. The bacterial population showed high growth dynamics in the middle range of mixing. It could be connected to the increased mass transfer during the mixing process [57]. It should be highlighted that increasing the velocity of the impeller may produce shear forces sufficient to disintegrate cells. That could be the case, notably when the speed exceeded 140 rpm (Re_mix_ = 6290). Aeration also played a crucial role in stimulating bacterial growth but with lesser impact. For the low range of impeller speeds, a high gas flow provided hydraulic mixing of fluid, improving the mass transfer process so that the growth rate increased [58].

The impact of mixing and aeration on bacterial cell viability, based on respiration in the resazurin assay, was also recorded during the processing time. The results obtained in these experiments are presented in Figure 5.

Based on the experimental results presented in Figure 5, bacterial cell viability strongly depended on the impeller speed. Mixing can create stress conditions for cells; thus, inhibited activity was observed for Re_mix_ = 2696 and 7189 relative to Re_mix_ = 0 (no mixing). The influence of aeration is weaker. However, the results indicated a high airflow rate (*V_g_* = 4.17 × 10^−6^ m^3^/s) and significantly improved metabolic activity after 12 h of processing compared to the ungassed conditions (*V_g_* = 0 m^3^/s). Oxygen is a vital electron acceptor for oxidative phosphorylation in bacterial metabolism. Even though *P. aeruginosa* can use nitrates to survive in anoxic conditions, this microorganism prefers aerobic conditions, and oxygen is essential for effective biomass production [49].

Additionally, the simultaneous impact of mixing and aeration on bacterial metabolic activity was studied in all tested setups after 6 and 12 h of the process. The results are illustrated in Figure 6.

The respiration level in a bacterial population grows together with the number of cells. Therefore, the resazurin assay is often used to estimate the number of cells in the sample [59]. The data distribution from this assay resembles that obtained with OD measurements. The recorded OD indicates growing biomass rather than a production of metabolites (or cell disruption) that could affect these measurements. Mixing at the middle range of impeller speeds (approximately Re_mix_ = 2600 to 3600) resulted in the best biomass production performance.

Similarly, aeration improved cell proliferation, especially for less intensive mixing (Re_mix_ up to 2600). Moreover, after 12 h of process viability, the highest air flow rate changes only slightly with the impeller speed up to 120 rpm (Re_mix_ = 5392). Afterwards, it harshly drops down, probably because of the increased shear stress. We assume that a high gas flow rate provided enough mixing to facilitate effective mass exchange in the system, providing the availability of nutrients and sufficient oxygen supply.

Pyocyanin quantification showed that the highest product concentration per milliliter of the culture was obtained when the highest impeller speed and aeration rate were applied (Figure 7a), which corresponds to relatively high oxygen mass transfer (k_L_a = 0.0013 s^−1^). In other setups, pyocyanin production was very low or negligible. Comparing the highest concentration obtained in this research to the results from the smaller scale experiments [60,61,62,63], the product concentration is relatively low, showing that the scale-up of pyocyanin production may be complicated and require a different setup than presented in this research. Research is still scarce on pyocyanin production on a scale larger than conical flask cultivation. 

The literature underlines that pyocyanin production is environmentally parameter-dependent [64]. Oxygen availability, temperature and pH can influence pyocyanin excretion. It is known that these factors can either alter the expression of genes or change the availability of substrates. Oxygen is crucial for pyocyanin production because it is required for the enzymatic conversion of 5-methylphenazine-1-carboxylic acid (one of the pyocyanin precursors) to pyocyanin [64]. The obtained results showed that the highest aeration rate and impeller speed led to the most efficient pyocyanin production, which is in line with the improved oxygen mass transfer. Moreover, it was previously shown that the low liquid volume ratio to the liquid free surface area could lead to higher pyocyanin production by *P. aeruginosa* ATCC^®^27853™ [60,61]. Nevertheless, the literature analysis proved that different *P. aeruginosa* strains may require different culturing conditions for pyocyanin production, e.g., temperature and agitation [65,66]. It is also worth mentioning that the most efficient pyocyanin production does not line up with the highest growth factor. The obtained data suggest that the bacterial cells focused metabolic pathways on biomass production instead of pyocyanin production for hydrodynamic conditions good for growth. Therefore, finding optimal production conditions is crucial to optimize this process.

Moreover, pyocyanin was reported to protect the bacterium from unfavorable conditions such as shear stress or chemicals [67]. Pyocyanin can promote the production of extracellular DNA that serves as, e.g., an additional scaffold in biofilms and strengthens their structure. This may be another justification for the most efficient pyocyanin production in the setup with the highest impeller speed and aeration rate.

Rhamphalolipid production, in contrast to pyocyanin production, was the most intense in setups providing high aeration or low impeller speed, which corresponds to moderate oxygen mass transfer (k_L_a = 0.0005 s^−1^). Interestingly, these results confirm that pyocyanin production could be inversely proportional to rhamnolipid excretion in the studied conditions (Figure 7b), which might be used to specifically design the process to produce either one metabolite or the other. It is worth emphasizing that rhamnolipid’s highest production is observed at the highest aeration, as in the case of pyocyanin, but in the absence of mixing. It may be related to the dissolved oxygen in the medium and k_L_a [62,68]. The decrease in rhamnolipid production with reduced airflow is most likely related to a decrease in dissolved air and, thus, a decrease in the expression of genes responsible for rhamnolipid production [68].

Figure 8 depicts data concerning pyocyanin (Figure 8a) and rhamnolipid production (Figure 8b), where the concentration of these metabolites is linked to biomass production and calculated as a yield factor. The results show that the highest yield factor for pyocyanin production was obtained for the highest gas flow rate and impeller speed. The highest yield factors for rhamnolipid production were calculated for low aeration rates and impeller speeds from medium to high.

## 4. Conclusions

Pyocyanin and rhamnolipids are valuable products of *P. aeruginosa* with versatile applications. However, laboratory bioreactors need to be tested to analyze the hydrodynamics they can create. In the current study, we tested a typical construction of a laboratory bioreactor with a 2 L working volume. Based on the CFD analysis we conducted (presented in Appendix A), in addition to the typical construction that involves a dual-impeller Rushton turbine and four rectangular baffles, the presence of other equipment, such as an internal cooler and pH probe, can significantly affect the hydrodynamics. Therefore, we tested and analyzed the fluid behavior for gassed and ungassed conditions in the form of typical parameters such as mixing time or power consumption. We proposed criteria that can help compare various process conditions based on the mixing power. Furthermore, we tested the growth and metabolite production of *P. aeruginosa* under selected conditions. We found that the changes in mixing and aeration significantly affected bacterial behavior. For the highest oxygen mass transfer and mixing (in the studied range), we found the most intensive pyocyanin production, while the rhamnolipids were produced under low mixing conditions and moderate oxygen mass transfer. Additionally, we found that moderate stirring and aeration are favorable for bacterial multiplication. Moreover, influences on hydrodynamics can lead to very different results—a process focused on a single compound or biomass with relatively high or low yield. The obtained data strongly suggest that hydrodynamic analysis is crucial in bioprocessing and can be used to control the bioprocess output.

## Figures and Tables

**Figure 1 microorganisms-11-00088-f001:**
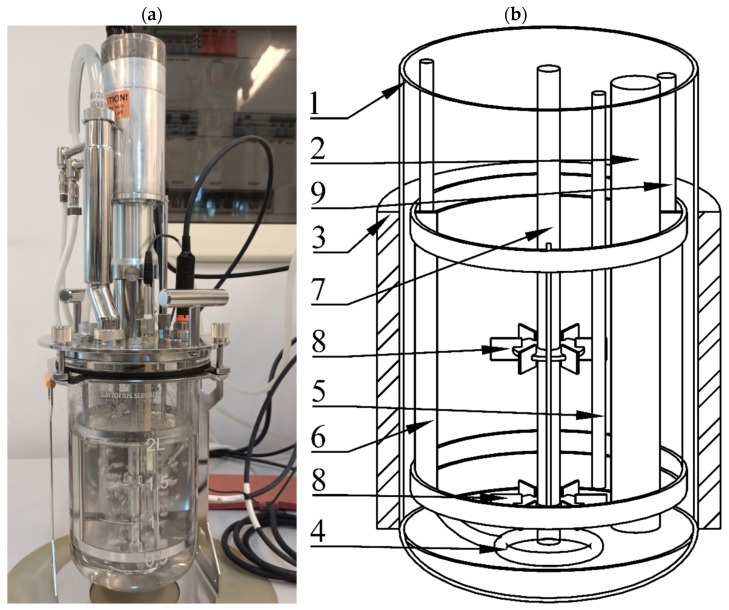
Experimental setup: (**a**) real view of the bioreactor, (**b**) bioreactor vessel scheme: 1–glass tank, 2–internal cooler, 3–external heating coat, 4–gas sparger, 5–temperature sensor, 6–baffles, 7–impeller shaft, 8–Rushton-type turbine, 9–injection port.

**Figure 2 microorganisms-11-00088-f002:**
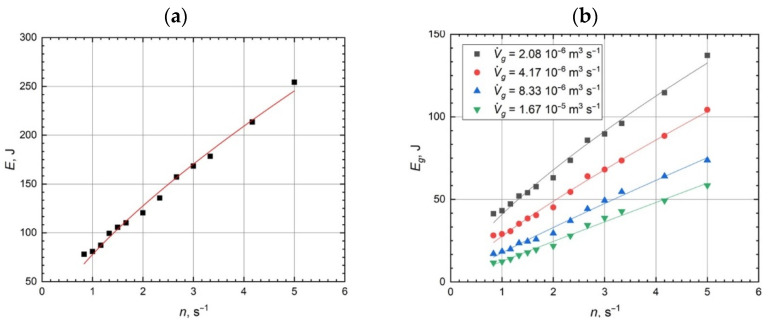
Mixing energy versus impeller speed: (**a**) ungassed conditions, (**b**) gassed conditions.

**Figure 3 microorganisms-11-00088-f003:**
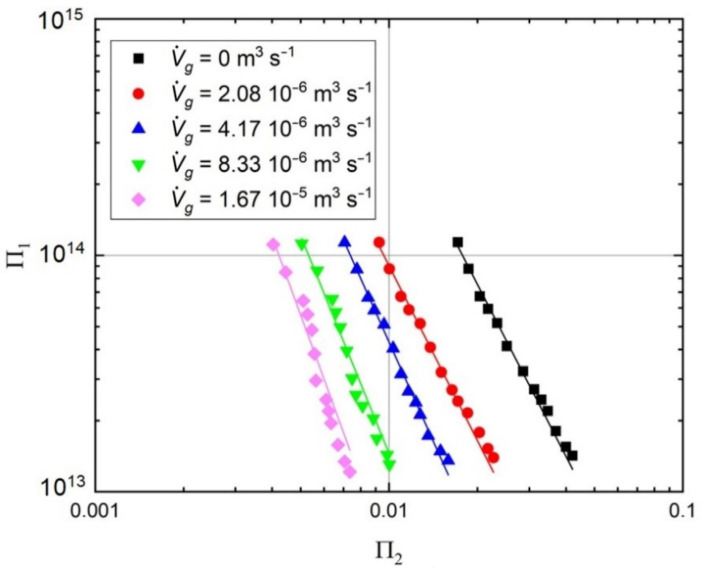
The dependence between the dimensionless numbers Π_1_ and Π_2_.

**Figure 4 microorganisms-11-00088-f004:**
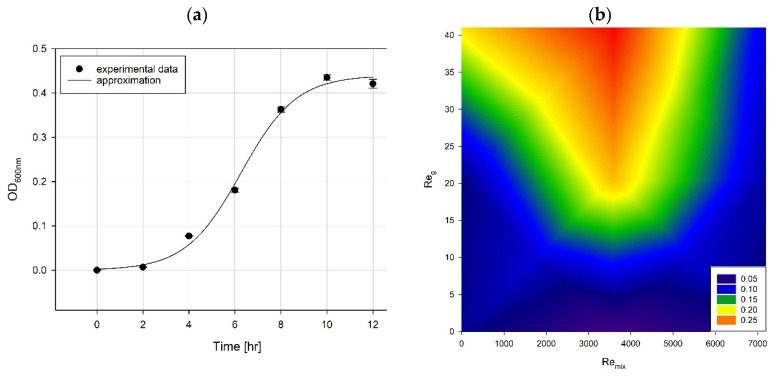
Typical growth curve of the studied bacterial cells (**a**) and bacterial growth factors for the studied process conditions (**b**).

**Figure 5 microorganisms-11-00088-f005:**
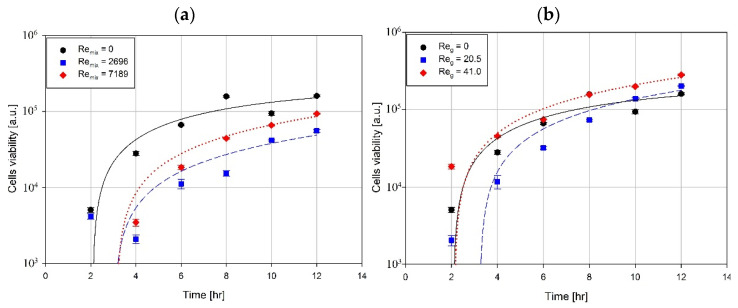
Changes in bacterial cell viability during the process: (**a**) impact of mixing speed, (**b**) impact of aeration.

**Figure 6 microorganisms-11-00088-f006:**
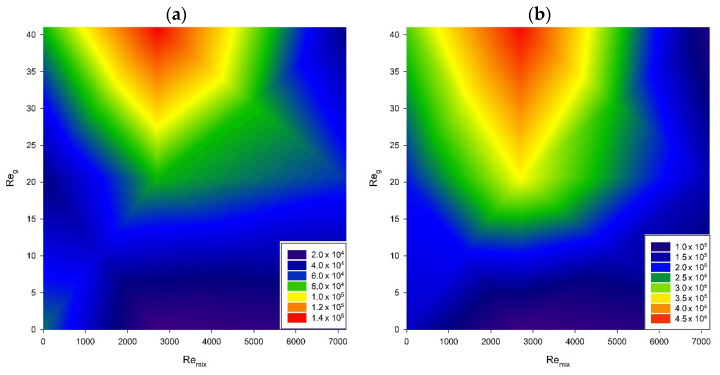
Impact of mixing with aeration on cell viability: (**a**) after 6 h, (**b**) after 12 h of process.

**Figure 7 microorganisms-11-00088-f007:**
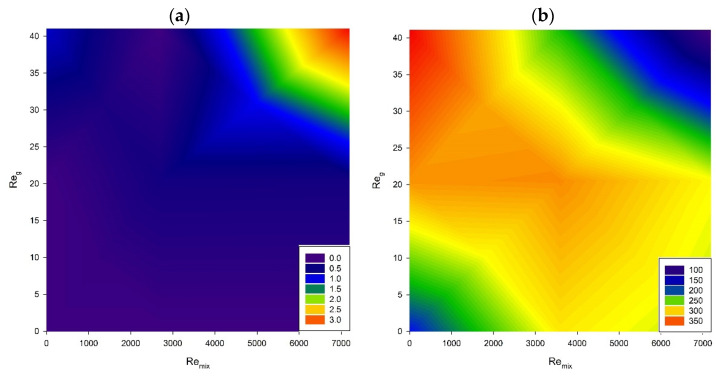
Contour plots of aeration and mixing impact on *P. aeruginosa* productivity: (**a**) pyocyanin (µg/mL), (**b**) rhamnolipids (µg/mL).

**Figure 8 microorganisms-11-00088-f008:**
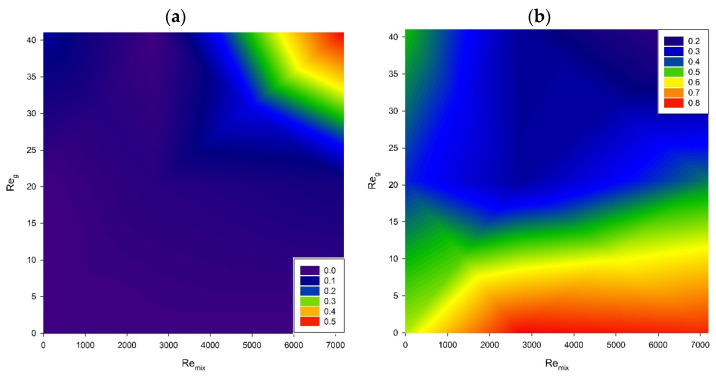
Impact of process conditions on the yield factor *Y_PX_* for (**a**) pyocyanin and (**b**) rhamnolipids.

## Data Availability

The data presented in this study are partially available in Appendix A. Additional data presented in this study are available on request from the corresponding author.

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
