# Peer review of "The Influence of Hydrodynamic Conditions in a Laboratory-Scale Bioreactor on Pseudomonas aeruginosa Metabolite Production"

_microorganisms, 2022, doi:10.3390/microorganisms11010088_

Round 1

Reviewer 1 Report

The authors investigated the influence of hydrodynamic conditions in a laboratory-scale bioreactor on the Pseudomonas aeruginosa metabolites production. However, there are some problems as following

1.       In the highlight section, I do not think the hydrodynamics study of a commercial bioreactor is a highlight.

2.       The abstract is too descriptive. Please add some quantities to it .

3.       As mentioned in this manuscript, the KLa is a very important parameter for Pseudomonas aeruginosa metabolites production. Please add the optimal value of  KLa for pyocyanin and rhamnolipid production.

4.       The ahthors cited 14 references(ref.16-29) about rhamnolipids production. But it is still not clear to me, what specific research questions the authors are asking and how they contribute to the existing study.

5.       I am wondering why the authors study the parameter of the power consumption. To my knowledge, this parameter is usually investigated in the stage of design and scale-up of bioreactor.

Reviewer 2 Report

It is irrelevant to highlight that commercial bioreactors are used, because it is indicated in the introduction, but it is not used in the discussion of results. The reactor configurations respond to a design aspect (geometry) rather than commercial availability. I suggest removing the commercial reactor part and focusing on the available designs of reactor and how the configuration affects the production of P. aeruginosa metabolites, because the authors do have experimental data. The introduction should be structured on the lack of information for the production of P. aeruginosa metabolites at the reactor level and how this approach can open up knowledge towards their industrial use.

Figure 5. Indicate bacterial metabolic activity but the authors not shown dates about metabolic process. Suggest indicate only cell viability.

Use the CFD data, to present the internal flow distributions (S6). These results support the effect of the reactor design on the obtained hydrodynamics.

The conclusions should emphasize the results obtained on the objective of the introduction. Remove line 318 because they do not present the data about this effect.

Reviewer 3 Report

Scaling up of the bioreactors are a well-known challenge and this paper presents important contribution for the bioreactor’s optimisation, useful for the industrial scale design. The work reports useful findings and directions for required bio-product where critical parameters that affect the performance were tested.

It has been reported that bio-products yield changes with the level of mixing with similar aeration. Failure in distributing oxygen uniformly to all the suspended microorganisms in the medium may lead to anaerobic side reactions that may spoil the whole batch.

Focus of the paper is to optimise the mixing work, however the sensitivity of microorganisms to the stress (that limits available high mixing rates) needs more information.  

The quality of publication can be increased additionally by including established theories on the bubble size that affect the gas-liquid mass transfer rates. Perhaps it would be worth tracing the bubble sizes in a function of mixing rate (and/or changed surface tension) during the fermentation in the transparent vessel. Oxygen transfer from the bubbles to the cells will be enhanced with the higher interfacial area g/l.  “a” gas-liquid surface area depends on the gas bubble sizes. Therefore, smaller bubbles create higher interfacial area and higher mass transfer rates. Explanation on how the bubble size can be varied and controlled – i.e. higher energy dissipation rates break the bubbles and surface tension is preserving larger sizes. The larger the ratio of Weber number to surface tension, the smaller the bubbles. Thus, obtained higher mass transfer rates from larger gas-liquid surface area that require high impeller speeds creates high stress and the microbial damage. In manuscript in lines 231-232 it has been estimated that it is 140rpm – was it done experimentally from the growth?  

It is important to justify the need for production of resistant pathogen (P. aeruginosa) and mention the toxicity of pyocyanin to inform potential use in food industry.

Minor corrections and considerations:

Figure 1 has numbers with the characteristics of the vessel without explanation. Is there a gas sparger nb 4? Rising bubbles will contribute to mixing.

Perhaps it’s worth adding modelling to abstract/intro as it’s randomly mentioned in 197 line stating briefly the purpose of doing it.

kLa – is this an error and “L” will be published as a subscript? Currently it’s confusing and stands as a separate symbol that supposed to mean sth.

Figure 4b – not clear what is this representing – what do the colors mean? (add in legend or in tittle)

Why are many results in the Supplementary Materials?

127 –  the meaning is not as meant to in “dissolved sensor”

136 – total plate count?

159-160 break into new line, fix the font179 – 180 break new line

179-181 break new line, font

 191-192 font changed

241 -242 – Not clear how the conclusion has been reached that activity is inhibited at high Re.

285-286 “previously shown that the low ratio of liquid volume to the interface area can lead to higher production of pyocyanin by P. aeruginosa ATCC®27853™ 70,71” – what does it mean? High gas hold-up? Small bubbles hence large gas-liquid interfacial area? Therefore high gas?

294 – how is this possible? “pyocyanin was reported to play 294 a role in protecting the bacterium from unfavourable conditions such as shear stress. Short explanation on the effect of shear stress on the bacterium cell is needed.

Interesting finding pyocyanin production could be inversely proportional to rhamnolipid excretion

Reviewer 4 Report

The paper under review deals with the research on the motion in a fluid flow inside a bioreactor. The authors characterized parameters such as mixing time, power consumption, and mass transfer in the 2L bioreactor. The results of the study are achieved via simulation and experiment. The authors show models and results of the simulations. The article tackles an important issue in reactors, heat/mass exchangers, and multiphase flow, and therefore is suitable for the journal. A structure of the paper is in accordance with principles of very good scientific reports. The paper is written in good English. The article contains adequate and appropriately selected 80 literature items.

In the opinion of the reviewer, the article can be accepted for publication.

Comments:

Ref. 1 should be corrected.

Eq. 1 should be corrected

Round 2

Reviewer 1 Report

The revised version is OK.